# Markov Transition Field Enhanced Deep Domain Adaptation Network for Milling Tool Condition Monitoring

**DOI:** 10.3390/mi13060873

**Published:** 2022-05-31

**Authors:** Wei Sun, Jie Zhou, Bintao Sun, Yuqing Zhou, Yongying Jiang

**Affiliations:** 1College of Mechanical and Electrical Engineering, Wenzhou University, Wenzhou 325035, China; Sunwei0977@163.com (W.S.); jzzzhou333@163.com (J.Z.); 00131041@wzu.edu.cn (B.S.); 2College of Mechanical and Electrical Engineering, Jiaxing Nanhu University, Jiaxing 314001, China

**Keywords:** tool condition monitoring, transfer learning, deep learning, domain adaptation, Markov transition field, variable conditions

## Abstract

Tool condition monitoring (TCM) is of great importance for improving the manufacturing efficiency and surface quality of workpieces. Data-driven machine learning methods are widely used in TCM and have achieved many good results. However, in actual industrial scenes, labeled data are not available in time in the target domain that significantly affect the performance of data-driven methods. To overcome this problem, a new TCM method combining the Markov transition field (MTF) and the deep domain adaptation network (DDAN) is proposed. A few vibration signals collected in the TCM experiments were represented in 2D images through MTF to enrich the features of the raw signals. The transferred ResNet50 was used to extract deep features of these 2D images. DDAN was employed to extract deep domain-invariant features between the source and target domains, in which the maximum mean discrepancy (MMD) is applied to measure the distance between two different distributions. TCM experiments show that the proposed method significantly outperforms the other three benchmark methods and is more robust under varying working conditions.

## 1. Introduction

The computerized numerical control (CNC) machine is a part of the important equipment in the advanced manufacturing industry. As one of the core components of the CNC machine, the cutting tool is the most vulnerable and wasteful component [1,2]. Along with the increasing wear of the tool, the cutting force, cutting heat, and cutting vibration increase significantly, which will lead to the decline of the workpiece’s surface quality. To achieve efficient machining processes, many researchers have conducted numerous studies on the wear mechanism of tools [3,4,5]. Zhang et al. [6] investigated diamond scratches during ultra-precision grinding. Wang et al. [7] used diamond tools to grind the cracks and studied the microstructure of the surface in detail. This method opens a new pathway to investigate the fundamental mechanisms of cutting [8]. In addition, a novel model of the maximum undeformed chip thickness is suggested for cutting, which is in good agreement with the experimental results [9]. However, the lack of timely tool change will affect the quality of the workpiece and even cause damage to the machine. Therefore, it is necessary to develop a reliable and robust tool condition monitoring (TCM) system to achieve timely tool replacement and make full use of the tool [10,11]. 

Since the late 1980s, TCM has been widely studied [12,13]. Since the mechanical equipment is becoming increasingly complicated, the traditional condition monitoring methods based on physical models and signal processing techniques have been less effective in TCM. With the great promotion of big data technology, data-driven methods have shown remarkable superiority in processing complex signals [14,15], which have also been introduced in TCM. For example, Yu et al. developed a novel approach based on the weighted hidden Markov model for tool remaining life prediction and tool wear monitoring [16]. Benkedjouh et al. proposed a nonlinear feature reduction and support vector regression for tool condition monitoring and remaining life prediction [17]. Prieto et al. proposed a deep neural network based on convolution long- and short-term memory (CLSTM) to predict vibration data of rotating machinery [18]. Mikolajczyk et al. presented a two-step method to automatically predict tool life in turning operations [19]. However, these data-driven methods need sufficiently labeled training samples to learn the model [20,21], which is difficult for TCM in the machining process due to the high cost of a lot of experiments [22]. The performance of data-driven methods could be poor with few labeled training samples [23,24]. To solve this problem, transfer learning (TL) has been developed with a small labeled sample in the target domain [25,26,27]. Li et al. proposed a partial domain adaptation method to achieve fault diagnosis [28]. Guo et al. developed a new intelligent method called the deep convolutional transfer learning network by using unlabeled data [29]. Chen et al. proposed a novel method for calibrating data labels using transfer learning algorithms that provides important insights into the application of unsupervised learning in wind turbine fault diagnosis [30]. Yang et al. proposed a feature-based transfer neural network (FTNN) that utilizes laboratory diagnostic knowledge to identify the health status of actual cases [31]. Marei et al. developed a convolutional neural network (CNN) method based on a transfer learning strategy to predict the tool conditions [32]. The above-mentioned works are helpful to build a transfer learning-supported TCM. However, in the actual industrial scene, there are missing categories in the target domain and varying working conditions, leading to the distributions between training data (source domain) and testing data (target domain) being significantly different, which dramatically lowers the performance of TL-based methods [33,34].

To overcome this problem, here, a new TCM method is proposed based on the Markov transition field (MTF) and the deep domain adaptation network (DDAN). MTF is employed to encode raw signals into 2D images according to the timeline, which can enrich the features of raw signals to assist the monitoring model in learning the condition pattern of tools. Transferred ResNet50 is employed to extract the deep features of these 2D images. Then, the extracted deep features of the source domain and target domain are adapted by the DDAN with maximum mean discrepancy to realize good performance under varying working conditions and missing categories. 

The rest of this paper is organized as follows. Section 2 introduces the theoretical background, Section 3 presents the proposed method, Section 4 discusses the experimental settings and the analysis of the results, and the conclusion is presented in Section 5.

## 2. Theoretical Background

### 2.1. Markov Transition Field

MTF was proposed by Wang and Oates in 2015 and encodes one-dimensional timeseries data into 2D images with time sequence [35]. 

An m-states Markov chain with states: $s_{1}$, … , $s_{m}$, can be represented by an *m*×*m* Markov transition matrix,$\;P_{mm}$, where $p_{ij}$ is the probability of state $s_{i}$ transiting to state $s_{j}$, $\sum_{i\;}p_{ij}=1$, and 1 ≤ i,j ≤ *m*, as shown in Equation (1):(1)$$P_{m\times m}=\left[\begin{array}{cccc} p_{11} & p_{12} & \cdots & p_{1m}\\ p_{21} & p_{22} & \cdots & p_{2m}\\ \vdots & \vdots & \ddots & \vdots \\ p_{m1} & p_{m2} & \cdots & p_{mm} \end{array}\right]$$

Given a timeseries X=(x1, x2,…, xn), a data point xt at time step *t* (1 ≤ *t* ≤ *n*) is first normalized using min–max normalization and scaled to between 0 and 1, as shown in Equation (2). Then, X is assigned to a corresponding state $s_{j}$ (or a quantile bin, $q_{j}$), where 1≤j≤m, and *m* is the number of states (or quantile bins).
(2)$$X=\frac{X-min(X)}{max(X)-min(X)}\;$$

In this way, an *m*×m Markov transition matrix, $P_{mm}$, associated with the timeseries *X* can be derived by first calculating $c_{ij}$ (1 ≤ i, j ≤ m), which is the count of data points in state $s_{i}$ transiting to state $s_{j}$. Afterwards, each entry, $p_{ij}$, of $P_{mm}$ can be derived as $p_{ij}$=$c_{ij}/\sum_{i\;}c_{ij}$. It can be easily checked that $\sum_{i\;}p_{ij}=1$. In practice, MTF captures the multi-span transition probability between any two data points in *X*, and constructs a transition matrix, $M_{n\times n}{=p}_{ij}(\;1\;\leq \;k,\;l\;\leq \;n,\;\mathrm{and}\;1\;\leq \;i,\;j\;\leq \;m)\;$, as shown in Equation (3), in which $p_{ij}$ is the probability that state $s_{i}$ of data point $x_{k}$ at time step k transits to state $s_{j}$ of data point $x_{l}$ at time step *l*. Compared with the Markov transition matrix, the MTF has extra-temporal information besides state transition possibilities. It is thus more suitable for representing and extracting features of timeseries. For a timeseries of a number, *n*, of data points, its associated MTF is an *n*×*n* matrix, which is usually regarded as an image to analyze and visualize. The characteristic representation flowchart of the one-dimensional timeseries processed by MTF is shown in Figure 1.
(3)$$M_{n\times n}=\left[\begin{array}{cccc} \left.p_{ij}\right|x_{1}\in s_{i},x_{1}\in s_{j} & \left.p_{ij}\right|x_{1}\in s_{i},x_{2}\in s_{j} & \cdots & \left.p_{ij}\right|x_{1}\in s_{i},x_{n}\in s_{j}\\ \left.p_{ij}\right|x_{2}\in s_{i},x_{1}\in s_{j} & \left.p_{ij}\right|x_{2}\in s_{i},x_{2}\in s_{j} & \cdots & \left.p_{ij}\right|x_{2}\in s_{i},x_{n}\in s_{j}\\ \vdots & \vdots & \ddots & \vdots \\ \left.p_{ij}\right|x_{n}\in s_{i},x_{1}\in s_{j} & \left.p_{ij}\right|x_{n}\in s_{i},x_{2}\in s_{j} & \cdots & \left.p_{ij}\right|x_{n}\in s_{i},x_{n}\in s_{j} \end{array}\right]$$

### 2.2. Domain Adaptation

Traditional machine learning algorithms perform poorly when the training and test data come from different distributions. In this case, domain adaptation becomes useful [36]. A domain consists of a data space, *x,* and a probability distribution, $P_{X}$, on its samples X ∈ x. Domain adaptation means to adapt useful knowledge from a source domain, S, to a target domain, T. Specifically, we are provided a source dataset (*X_S_*, *Y_S_*) = {$\;{(X}_{s}^{1},Y_{s}^{1}),$ ${(X}_{s}^{2},Y_{s}^{2}),$ ${(X}_{s}^{3},Y_{s}^{3}),$ … , ($X_{s}^{m},Y_{s}^{m}$)} and a target dataset with limited unlabeled data (*X_T_*) ={ ${(X}_{t}^{1}),$ ${(X}_{t}^{2}),$ … , ${(X}_{t}^{n})$} for model building. The trained model is expected to demonstrate great classification or regression accuracy on new target domain samples. Figure 2 illustrates the basic problem of domain adaptation.

## 3. Proposed Method

### 3.1. Framework MTF-DDAN

The framework of the proposed TCM method based on the MTF and DDAN (MTF-DDAN) is shown in Figure 3, including feature representation, feature extraction, domain adaptation, and classification. In the proposed MTF-DDAN method, MTF is employed to represent the implicit features of the original timeseries and ResNet50 as a feature extractor to extract the high-dimensional features (*H*) in the samples. The DDAN model objective function consists of the following two parts: 

(1) The basic classification loss ($L_{clc}$) for source supervised learning. This part uses a typical supervised learning scheme for source domain data only and employs a cross-entropy loss function. 

(2) The maximum mean discrepancy (MMD) loss ($L_{mmd}$) between the distributions of *D_s_* and *D_t_*. MMD is a method of distance measurement which measures the distance between two different but similar distributions in the reproducible kernel Hilbert space, $H_{k}$(RKHS). It is a kernel learning method that maps the original variables into the RKHS space. MMD can be estimated using Equation (4):(4)$$MMD_{k}^{2}(D_{s},D_{t})={\left\|\frac{1}{m}\sum_{i=1}^{m}\varphi (x_{i}^{s})-\frac{1}{n}\sum_{j=1}^{n}\varphi (x_{j}^{t})\right\|}_{{}_{H_{k}}}^{2}$$

The MMD value is expected to be a small quantity if the distributions of *D_s_* and *D_t_* are similar, where ϕ(·) is the nonlinear mapping from the original feature space to RKHS, representing the features extracted by the deep model. The inner product in the RKHS space can be converted into a kernel function, so MMD can be calculated directly from the kernel function, as shown in Equation (5):(5)$$MMD_{k}^{2}(D_{s},D_{t})=\frac{1}{m^{2}}\sum_{i=1}^{m}\sum_{j=1}^{m}k(x_{i}^{s},x_{j}^{s})-\frac{1}{mn}\sum_{i=1}^{m}\sum_{j=1}^{n}k(x_{i}^{s},x_{j}^{t})+\frac{1}{n^{2}}\sum_{i=1}^{n}\sum_{j=1}^{n}k(x_{i}^{t},x_{j}^{t})$$
where k(·,·) is a kernel function. Here, the Gaussian kernel function is used as the characteristic kernel. Two different domain features would be drawn closer in $H_{k}$ by minimizing Equation (5).

### 3.2. MTF-DDAN Model Training Phase

Due to the new processing conditions, it is difficult to obtain labeled data in time, and only unlabeled data can be obtained in the milling process and cannot be effectively learned using traditional deep learning methods. Firstly, the raw signals collected from the experiment could be represented in 2D images through MTF as training samples. Secondly, ResNet50 pre-trained on ImageNet [37] was chosen to extract deep features from the source and target domains. After feature extraction, DDAN can achieve domain-invariant knowledge learning by calculating MMD between two distributions in RKHS, automatically. It achieves the DDAN by using the Adam optimizer through the back-propagation algorithm.

In the model testing stage, timeseries signals collected by an acceleration sensor were characterized by MTF and input into the trained model for condition prediction. The model parameters are shown in Table 1.

## 4. Experiment Investigation

### 4.1. Experimental Setup

The milling experiments were carried out on a CNC milling machine (DMTG VDL850A), and AISI 1045 steel was used as the workpiece material with L300 mm × W100 mm × H80 mm. The cutting tool is a three-flute uncoated carbide end-milling cutter with a 10 mm diameter. Milling experiments were tested via dry milling. Experimental settings are shown in Figure 4. 

The experiment implemented a total of three tools to simulate the complex working conditions in the machining process. The parameters of each experimental procedure are variable, and D1, D2, and D3 were combined to construct the source domain dataset and the target domain dataset. Experiment parameters are displayed in Table 2.

Vibration signals are mainly caused by dynamic components in the cutting force, which are closely related to the dynamic characteristics of the whole cutting system [38,39]. It often contains the most abundant condition information in the process of machining. Compared with a dynamometer, a single acceleration sensor has the advantages of low cost, convenient installation, and does not affect normal machining. The acceleration sensor was installed on the lower surface of the workpiece, and data were collected by employing a signal acquisition device (ECON dynamic signal analyzer, as shown in Figure 4b). The sampling frequency was set to 12,000 Hz. Tool wear was measured using a tool microscope (GP-300C) containing a CCD camera (Figure 4d). The end-milling cutter was placed vertically under the microscope to measure the wear length of each blade. Figure 5 shows the time domain signals of three milling tools, where each tool is plotted with three different wear states, from top to bottom showing slight wear, stable wear, and sharp wear, respectively. In general, there were differences in the signals with different cutting parameters. It can be seen from Figure 5 that this difference was more obvious when the spindle speed was not the same, and this case tended to have a negative transfer phenomenon, which made the next experiments challenging.

According to ISO3685-1977, the tool wear is defined as the wear width VB on the side of the tool, but since the variation of the wear width on the side is not obvious enough during the actual experiment, which can easily lead to measurement errors, we chose the maximum wear length on the end face of the tool as the wear standard, VB = Max (VB1, VB2, VB3). Since the tool and the workpiece were dry-cut during our experiments, the tool wear was relatively fast, and the offline measurement of tool wear was performed once for every cutting of the same feed length of the milling cutter. Our milling cutter reached the end of life after ten measurements, so we only divided the dataset into three wear states: slight wear (VB ≤ 0.8 mm), stable wear (VB = 0.8~1.6 mm), and sharp wear (VB ≥ 1.6 mm). Figure 6 shows the tool wear after 1, 5, and 10 finishes on a single workpiece surface. Figure 7 shows the change in the wear value of the experimental tool at each downtime measurement. 

### 4.2. Datasets’ Description

Six transfer tasks were designed to verify the effectiveness of MTF-DDAN, including task 1 (D1→D2), task 2 (D2→D1), task 3 (D1→D3), task 4 (D3→D1), task 5 (D2→D3), and task 6 (D3→D2). The left and right sides of “→” denote the source and target samples, respectively, and the working conditions between the three domains were different. The source domain dataset in each transfer task during the experiment was labeled during the training process, and the target domain was unlabeled. In the experiments, the sample size of both the source and target domain datasets was 300. Each sample contained 3000 data points of timeseries and was converted into 2D images with a size of 224 × 224 through MTF.

### 4.3. Results and Discussion

To verify the sensitivity of the experimental data to tool wear, we decomposed the original signal into different frequency bands using wavelet packet decomposition, and then analyzed the reconstructed signal for each band. Figure 8 shows the performance of MTF-DDAN using different frequency band data. It can be seen that the important information related to tool wear was mainly distributed in the low-frequency part of the frequency range (0~1500 Hz). Therefore, we used reconstructed signals in the (0~1500 Hz) frequency band for different transfer task experiments.

The prediction results of DDAN and three benchmark methods (AlexNet, ResNet, and DAAN) on the six TCM transfer tasks are shown in Table 3 and Figure 9. The results of each transfer task are the average of five times. From the average classification accuracy of the six transfer tasks, it can be seen that the method proposed in this paper had a more stable performance and can realize the condition monitoring of the tool in the case of variable working conditions. Since there were missing labeled target data, the AlexNet and ResNet50 methods were not adapted to train directly on unlabeled data and could only rely on labeled source domain data. The DANN (dynamic adversarial adaptation network) is an adversarial learning transfer method. It can be seen that DDAN significantly outperformed the other methods in most transfer tasks and achieved comparable performance in task 4. D1 and D3 had great differences in cutting parameters: only the feed speed was the same, which led to the poor performance in transfer tasks 3 and 4 compared to other tasks. Transfer tasks 1 and 2 still showed more than 85% classification accuracy in the case of large differences in cutting parameters, which was at least 4.47% higher than the other three comparison methods. The classification accuracy improvement shows that our proposed MTF-DDAN method can achieve better performance across different working conditions. 

Figure 10 reflects the average standard deviation of each method for the six transfer tasks, and it can be seen that the traditional deep learning methods using only source domain data for training could not achieve better classification results in the target domain and were less stable. The domain adaptation methods represented by DAAN and DDAN had better results for tool condition monitoring under variable working conditions, and their stability had obvious advantages compared with other methods. In general, our proposed DDAN can achieve better classification performance and stability under variable working conditions.

## 5. Conclusions

In this paper, a new TCM method combined with the Markov transition field and the deep domain adaptation network (MTF-DDAN) was proposed. The experimental results showed that the generic information related to tool wear under different working conditions was mainly contained in the (0~1500 Hz) frequency band, which is helpful for the following tool condition monitoring under variable working conditions. Six transfer tasks’ result showed that the proposed method significantly outperformed three other benchmark methods, whereby the average classification accuracy of the proposed method was at least 8% higher than that of the other methods. Therefore, the proposed method in this paper holds the promise of being effectively applied to realistic machining processes to accurately identify the wear condition of tools so that they can be replaced in time. Furthermore, to improve the prediction accuracy of TCM, the kernel function and its parameter of MMD could be optimized by data-driven methods, and the MTF could be improved in the time and frequency aspects.

## Figures and Tables

**Figure 1 micromachines-13-00873-f001:**
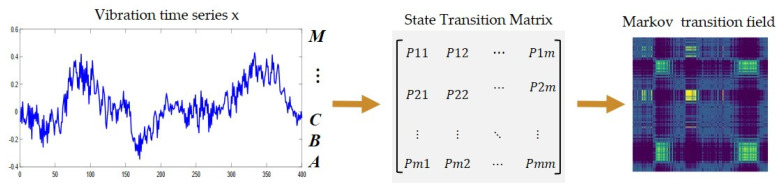
Timeseries feature representation by MTF.

**Figure 2 micromachines-13-00873-f002:**
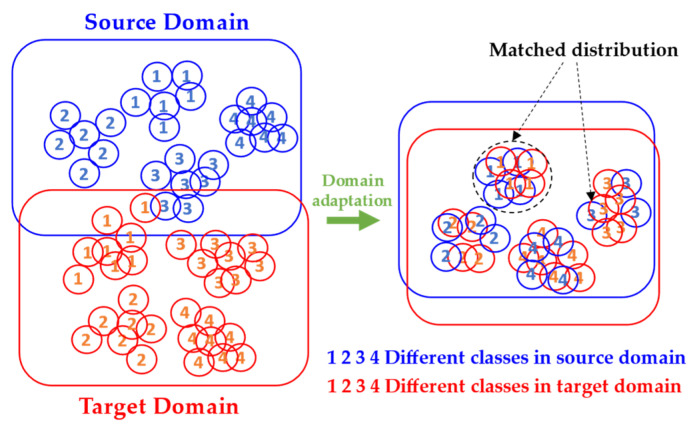
Basic principle of domain adaptation.

**Figure 3 micromachines-13-00873-f003:**
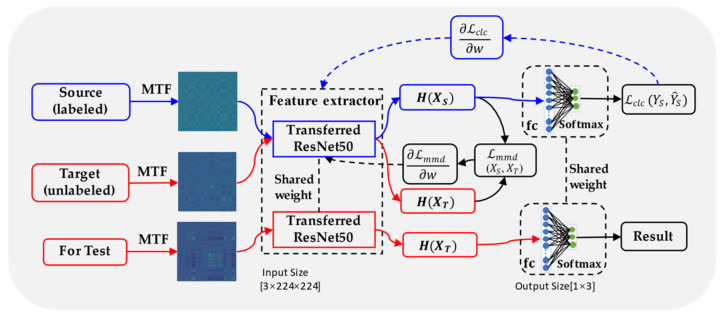
The MTF-DDAN framework. *H* represents the high-dimensional features extracted by the deep network, $L_{clc}$($Y_{S}{,\;\hat{Y}}_{S}$) refers to the classification loss function, $Y_{S}{,\;\hat{Y}}_{S}$ represent source label and predicted label, respectively, $L_{mmd}$ ($X_{S},X_{T}$) stands for MMD loss function, and $X_{S}\;\mathrm{and}\;X_{T}$ represent source and target data, respectively.

**Figure 4 micromachines-13-00873-f004:**
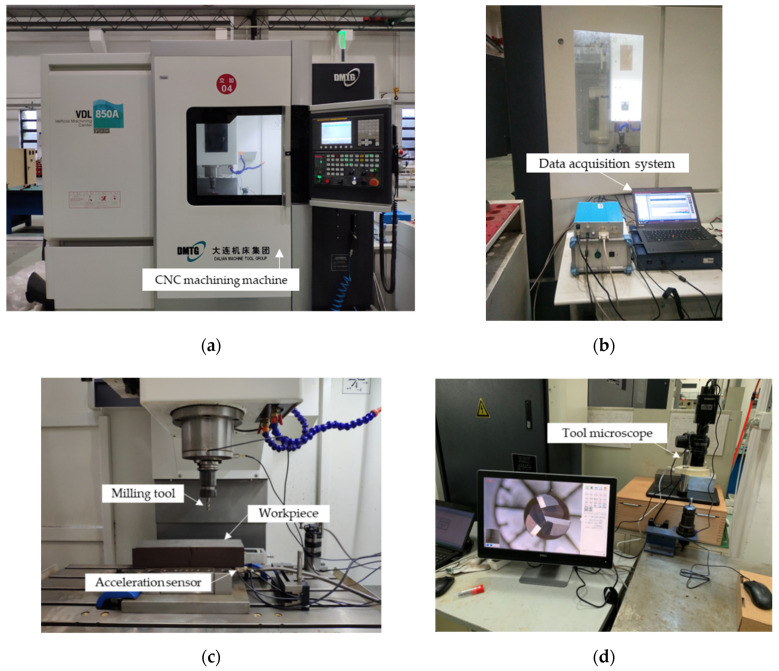
The experimental setting. (**a**) CNC machine, (**b**) data acquisition system, (**c**) experimental platform, and (**d**) tool microscope.

**Figure 5 micromachines-13-00873-f005:**
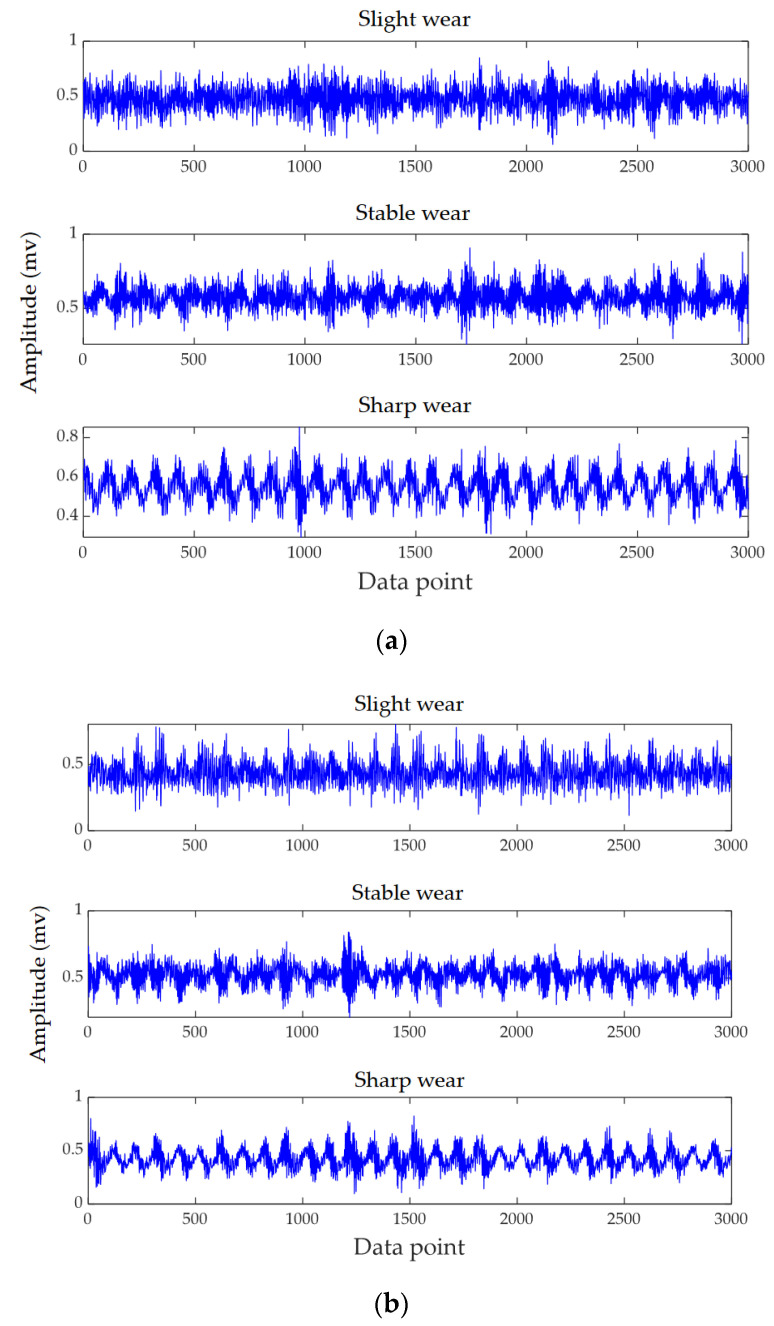
Time domain signals in different wear conditions. (**a**) Tool 1, (**b**) Tool 2, and (**c**) Tool 3.

**Figure 6 micromachines-13-00873-f006:**
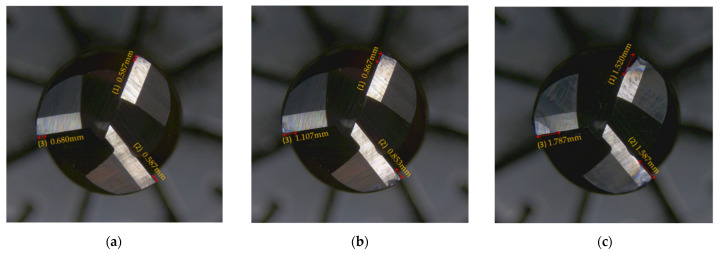
Tool wear images of different tooling processes. (**a**) Wear of first tooling process, (**b**) wear of fifth tooling process, and (**c**) wear of tenth tooling process.

**Figure 7 micromachines-13-00873-f007:**
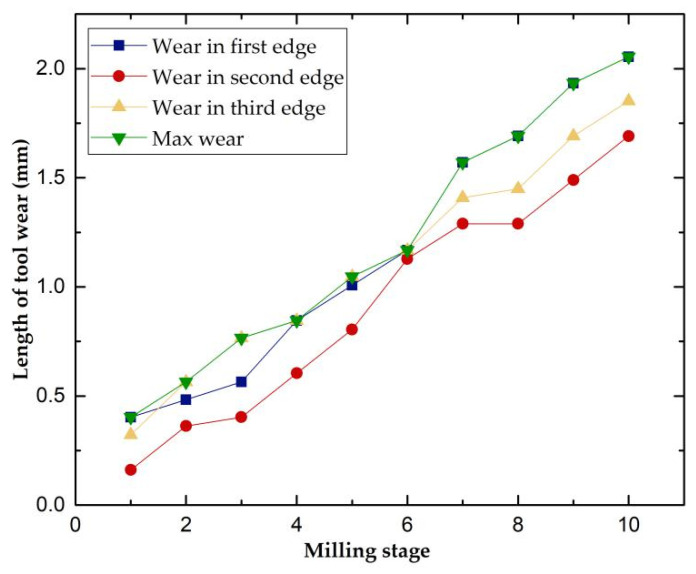
Evolution of tool wear.

**Figure 8 micromachines-13-00873-f008:**
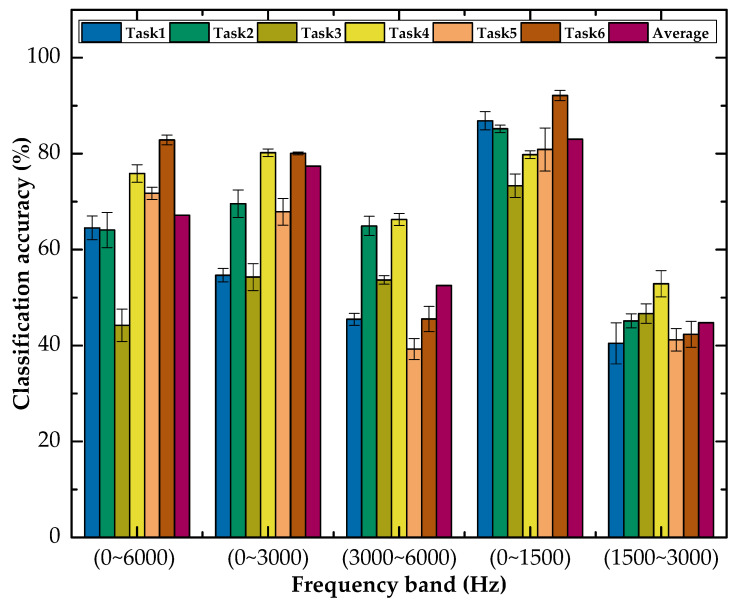
Classification accuracy (%) on MTF-DDAN in different frequency bands.

**Figure 9 micromachines-13-00873-f009:**
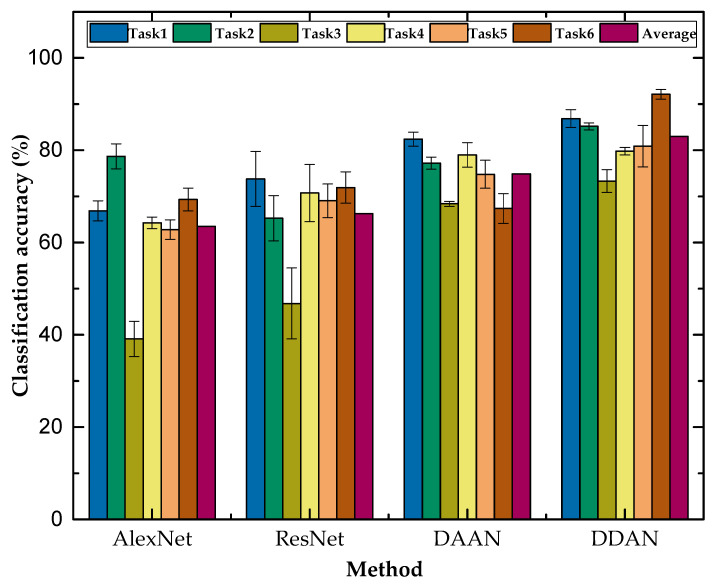
Classification accuracy (%) on the TCM dataset with six transfer tasks.

**Figure 10 micromachines-13-00873-f010:**
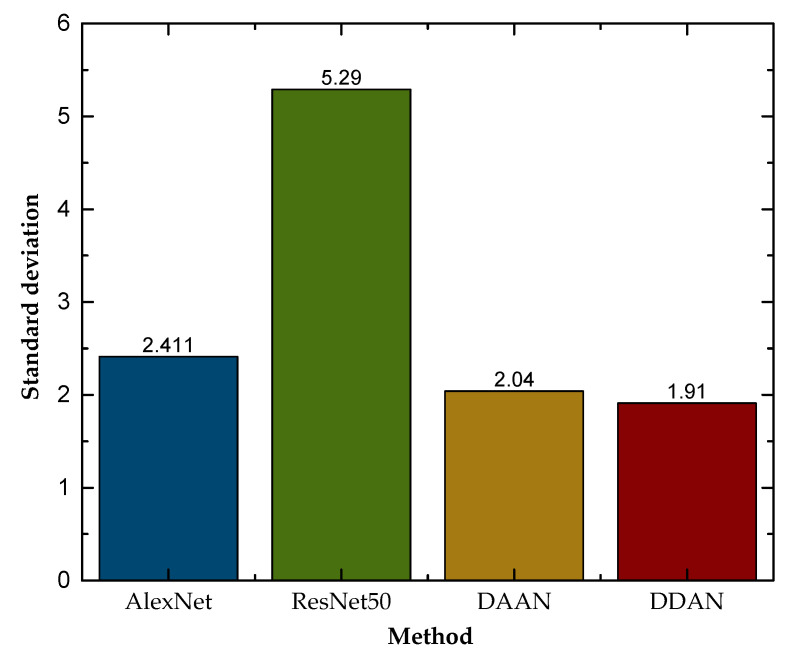
Standard deviation of different methods.

**Table 1 micromachines-13-00873-t001:** Model parameters.

Parameter	Image Size	Learning Rate	Dropout	Batch Size	Optimizer	Loss Function
Value	224×224×3	5×e^−4^	0.5	8	Adam	Cross-Entropy Loss

**Table 2 micromachines-13-00873-t002:** Experimental parameters.

Domain	Tool Number	Spindle Speed (rpm)	Feed Speed (mm/min)	Axial Cut Depth (mm)
D1	1	2300	400	0.4
D2	2	2400	500	0.5
D3	3	2400	400	0.5

**Table 3 micromachines-13-00873-t003:** Classification accuracy (%) on the TCM dataset with six transfer tasks.

Method	D1→D2	D2→D1	D1→D3	D3→D1	D2→D3	D3→D2	Average
AlexNet	66.87	78.67	39.12	64.27	62.80	69.33	63.51
ResNet50	73.79	65.27	46.80	70.73	69.07	71.93	66.27
DAAN	82.40	77.20	68.40	79.00	74.80	67.40	74.87
DDAN	86.87	85.20	73.33	79.80	80.87	92.13	83.03

## Data Availability

Not applicable.

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
