# Peer review of "Markov Transition Field Enhanced Deep Domain Adaptation Network for Milling Tool Condition Monitoring"

_micromachines, 2022, doi:10.3390/mi13060873_

Round 1
Reviewer 1 Report
The paper is devoted to the issues of tool condition monitoring (TCM) in the milling process.
The title corresponds to the content, the order and structure of the paper is correct.
Reviewer comments:
1. The scope of the paper only partially relates to the journal profile.
2. The authors present an MTF-DAN framework that is very advanced conceptually and computationally. Without detracting from their work, it should be noted that the model is not well presented, lacking labels and acronyms. There is no indication of why such an elaborate network was used to identify excessive wear on a tool that should have been replaced long ago.
The reviewer has doubts about the unambiguity of marking and assessment of wear according to the ISO standard. The authors themselves do not use unambiguous designations - Fig. 6, 7 and text.
Discussion of the results and conclusions should be complemented and justified more fully. There is lack of unambiguous indication of what is input and output in the model. There is a lack of showing raw data, and fig. 7 with labeling "Milling process" is highly misleading.
Very interesting article but needs improvement
Reviewer 2 Report
Tool condition monitoring is a hot topic that has been studied in recent years. A TCM method combining Markov transition field (MTF) and domain adaptation network (DAN) is proposed in this paper. Although the research results obtained in this paper may have certain significance, in the reviewer's view, it lacks innovation and needs further thorough revision, and the quality of the paper needs to be improved. The method used has been reported in many literatures. Some of the comments below may be of some help in improving the work of this paper:
(1) In the introduction, there is too little discussion about the research significance and purpose of the paper, and it is not deep enough. The elaboration and discussion of the existing literature is relatively simple.
(2) In the second part of this paper, the proportion of the content explained by each theoretical method should be adjusted appropriately. The structure and advantages of the proposed method should be elaborated. Too much information is included in Figure 4, and the quality of the figure should be appropriately improved.
(3) Some information in Figure 5 should be explained and labeled, otherwise it will be difficult for readers to understand. Furthermore, the quality of Figure 6 needs to be improved.
(4) The conclusions of this paper require a thorough revision. Each research content needs to be described in detail item by item.
Reviewer 3 Report
This manuscript proposes a method of tool condition monitoring for milling. Follows are the comments on this manuscript.
- The label of references is not correct, which should comply with the requirements of journal.
- In this manuscript, cutting tool, force, machining, method and approach are addressed, while recent significant progresses on them are lost. In this regard, follows should be added in Introduction. A novel method is significant for cutting and high performance manufacturing (INTERNATIONAL JOURNAL OF PRECISION ENGINEERING AND MANUFACTURING-GREEN TECHNOLOGY 8 (2021) 193-204; INTERNATIONAL JOURNAL OF ADVANCED MANUFACTURING TECHNOLOGY 94 (2018) 855-865; CIRP ANNALS-MANUFACTURING TECHNOLOGY 57 (2008) 676-696; PROCEEDINGS OF THE INSTITUTION OF MECHANICAL ENGINEERS PART B-JOURNAL OF ENGINEERING MANUFACTURE 228 (2014) 527-539). A novel approach of single grain grinding is proposed at 40.2 m/s and nanoscale depth of cut, in which the speeds used are four to seven orders of magnitude higher than those employed in nanoscratching (CIRP Annals-Manufacturing Technology, 64 (2015) 349-352). Force, stress, depth of cut and size of plastic deformation are calculated at the onset of cutting (Journal of Manufacturing Processes, 75676-696 (2022) 617-626). This method opens a new pathway to investigate the fundamental mechanisms of cutting (Nano Letters 18 (2018) 4611-4617). In addition, a novel model of the maximum undeformed chip thickness is suggested for cutting, which is in good agreement with experimental results (Scripta Materialia, 67(7-8), (2012) 657-660; Scripta Materialia, 67(2), (2012) 197-200). Under the breakthrough of theories, novel cutting approaches are developed, which is a great contribution to the traditional cutting and manufacturing (Journal of Alloys and Compounds, 726 (2017) 514-524).
- In Fig. 5, description on components of setups should be added in photographs for understanding.
- In Fig. 6, the digits and words are too small to see, which should be enlarged as that in text. Additionally, signs of (a), (b), (c) are lost.
- Predicted and experimental results should be illustrated in an image for comparison. The gap between the two results should be elucidated clearly.
- For a test, at least three experiments should be performed, and then an average value and standard deviation could be obtained. Once result is not convinced for a qualified test.
- The model suggested in this work should be compared with those published previously, and the advantages should be highlighted. Of course, the disadvantages should also be mentioned.
Round 2
Reviewer 3 Report
All the comments and concerns have been addressed and corrected, and I think it is appropriate for publication.